| Open Peer Review | Genetics and Molecular Biology | Minireview

# The best from both disciplines: integrating human and microbial signatures from whole genome sequencing to advance cancer diagnostics

Tina Moser,[1] Matthias J. Moser,[1] Alexander Mahnert[2]

**ABSTRACT** Liquid biopsies are transforming oncology, enabling earlier diagnosis, dynamic treatment guidance, and personalized precision medicine, yet current approaches focusing mainly on circulating host cell-free DNA (cfDNA) neglect crucial information within co-existing microbial cell-free DNA (mcfDNA). This review argues for the combined potential of simultaneously analyzing host and microbial signals from samples like blood, specifically focusing on circulating tumor DNA (ctDNA) as the key host component. While ctDNA analysis is already used to guide treatment decisions, the detection of mcfDNA—although present in smaller amounts compared to total cfDNA—offers a distinct and complementary opportunity to identify disease-causing microbes and investigate the host-associated microbiome in the context of cancer. Leveraging machine learning strategies is essential to integrate these multi-view data sets and realize their full potential for enhancing liquid biopsy applications, particularly in early cancer detection.

## UNLOCKING MICROBIAL SIGNATURES FOR DIAGNOSTIC INNOVATION

The human microbiome is commonly perceived as our second genome (1) due to its genetic diversity, versatility, importance for immune modulation, and human metabolism (2). Human evolution is taking place in a microbial world (3). Microbes educate our immune system and occupy niches that could be exploited by pathogens (4). Microbes accompany human beings for their entire life and are omnipresent during disease, treatment, and until death. Hence, microbial traces or signatures can be powerful biomarkers to assess the medical conditions of the human body, and even predict the course of disease and success of a therapy (5–10). Our understanding of the human microbiome was mainly driven by investigating stool samples and applying next-generation sequencing (NGS). Using stool samples has the advantage of providing high microbial biomass and low human background noise for downstream sequencing analysis. Nevertheless, it remains questionable whether stool samples are representative of the entire gastrointestinal tract and its complex dynamics (11). Generally, diagnostics in the medical field often rely on sample types other than human stool alone—for instance, blood, urine, or biopsy samples. It is therefore not surprising that these alternative sample types have already been used for microbiome profiling and microbiome-informed biomarker searches for potential diagnostic applications (12). Still, the differentiation between correlation and causality—a common challenge in microbiome research—is evident across alternative sample types. For instance, while viruses like papillomavirus for cervical cancer (13) or bacterial pathogens like *Helicobacter pylori* (Cag-A) for stomach cancer (14) and *Fusobacterium nucleatum* (adhesin A, autotransporter protein 2) for colon cancer (15) have been widely accepted as causative agents of cancer, other signals of the microbiome are still part of a lively debate. Just recently, Gihawi et al. (16) questioned the cancer-type-specific microbial signatures

Address correspondence to Alexander Mahnert, alexander.mahnert@medunigraz.at.

The authors declare no conflict of interest.

See the funding table on p. 10.

reported by Poore and colleagues (12) and attributed many observations and results to bioinformatic problems, including flawed data, problematic machine learning (ML), normalization artifacts, and database contamination. Meanwhile, Sepich-Poore et al. retracted their original manuscript from 2020 but reaffirmed the robustness of cancer-associated microbiome signals using updated bioinformatic workflows that included batch correction, improved host read depletion with a newer human genome reference, cross-method consistency analyses, and additional validation steps (17). Irrespective of one's angle on this ongoing debate, the scientific debate emphasizes a common challenge for microbiome research in low-biomass environments: the need for rigorous controls, specific adaptation, and standardized bioinformatic workflows, and the difficulty of distinguishing correlations from causal relationships. Hence, in addition to these manifold technical hurdles, exploiting genetic traces of the microbiome through whole-genome sequencing (WGS) remains a promising target, especially in a joint multi-view approach for modern cancer diagnostics (Box 1) .

In this minireview, we aim to address the potential of liquid biopsy (LB) to simultaneously extract tumor-specific alterations and microbial signatures to enhance cancer diagnostics. We will cover how these signals can be extracted using a single assay (i.e., WGS), the methodological limitations and challenges, and strategies for integrating these signals using ML to harness the potential of a multi-view approach .

## THE ROLE OF THE MICROBIOME IN CANCER

The impact of the microbiome within our body was long overlooked. However, today, it is a widely discussed topic that extends its influence to various aspects such as aging, physical health, and overall well-being. Influenced by factors like sex, age, lifestyle, and diet, the trillions of microbes in the human body—including bacteria, archaea, viruses, and eukaryotes like fungi and protozoans—play a crucial role in maintaining our health through a symbiotic relationship with the human body (18–20). Most of the human microbiome's biomass is located in the gastrointestinal tract, especially in the colon, i.e., the gut microbiome (21). Changes in the composition of these microbes have been linked to numerous diseases, including depression, diabetes, inflammatory bowel disease, and autoimmune disorders (22).

In oncology, studies over the past decade have begun to uncover intricate links between microbes and cancer (23–27). More recently, the gut microbiome has drawn increasing attention in the era of precision oncology, as gut dysbiosis, a condition characterized by an imbalance in the types of microbes in the gut, contributes substantially to gastrointestinal cancers by causing, modulating, and affecting the progression and therapy response (28). In addition to the importance of the gut microbiome in cancer, intra-tumoral microbial profiles may provide valuable insights into cancerous diseases. It has been reported that microbial profiles not only differ between primary tumors and non-malignant tissues but also among different tumor entities, each exhibiting unique intra-tumoral microbial fingerprints (12, 29). Nejman et al. (29) demonstrated that the bacterial composition varies across seven cancer types, including breast, lung, ovary, pancreas, melanoma, bone, and brain tumors. Moreover, the research by Narunsky-Haziza and colleagues (30) has revealed that despite low levels of fungal DNA across many major human cancers, there are cancer-type-specific differences in the composition of fungal ecologies.

Recent data show that differences in gut and intra-tumoral microbe populations may lead to cancer progression and discrepancies in the efficacy of cancer therapies (28, 31–33), including chemotherapy, radiotherapy, immunotherapy, and surgery, between patients. For example, patients harboring specific gut microbe taxa were associated with better responses to immunotherapy treatment across different tumor types (28). In addition to the gut microbiome, altered intra-tumoral microbiota are associated with resistance to cancer treatment (31). Battaglia et al. (34) highlighted that lung cancer patients whose metastases contain *Fusobacterium* tend to show resistance to immune checkpoint blockade treatment compared to patients without

---

**BOX 1. GLOSSARY**

**Feature**

A specific, measurable characteristic of the data that serves as input for a machine learning model. For example, a feature might represent the cfDNA fragment length or the abundance of a specific microbe.

**Modality**

A distinct category or source of data. Therefore, a multimodal analysis integrates different data types, such as cfDNA fragmentomics and microbial composition data, to form a more comprehensive view.

**Autoencoder**

An unsupervised neural network that learns to efficiently encode data into a compressed format and then decode it back to its original form. This process forces it to identify the most critical patterns in the data.

**Penalized logistic regression**

A classification method that adds a regularization term to the loss function to prevent overfitting by shrinking model coefficients, commonly using L1 (lasso) or L2 (ridge) penalties.

**Multidimensional scaling (MDS)**

A statistical technique that visualizes the similarity or dissimilarity between data points by placing them in a low-dimensional space while preserving their pairwise distances as accurately as possible.

**Support vector machine (SVM)**

A supervised learning algorithm that finds the optimal hyperplane to separate data into distinct classes by maximizing the margin between them.

**Meta-learner**

A higher-level model in machine learning that learns how to combine or optimize the performance of multiple base models, often used in ensemble methods or meta-learning frameworks.

**Classifier**

A machine learning model that assigns input data to predefined categories or classes based on learned patterns from labeled training data.

**Regressor**

A machine learning model that predicts continuous numerical values based on input features by learning patterns from labeled training data.

**Neural networks**

Computational models inspired by the structure and function of biological neurons that learn patterns from data to perform tasks like classification, prediction, and pattern recognition.

**Graph convolutional neural network (GCN)**

A deep learning architecture that generalizes convolution operations to graph-structured data, enabling the extraction of node features by aggregating information from neighboring nodes.

---

this bacterium. Furthermore, intra-tumoral bacteria can influence tumor biology through multiple, conserved mechanisms across cancer types. For instance, certain bacterial genotoxins, such as colibactin produced by *Escherichia coli* and cytolethal distending

toxin from *Campylobacter* spp., can directly induce DNA double-strand breaks and genomic instability, thereby promoting mutagenesis and tumor evolution (35). Bacterial metabolites and cell wall components can also alter chromatin accessibility, modulate histone modifications, and reshape DNA methylation patterns, ultimately reprogramming oncogenic signaling and stress-response pathways (35, 36). Moreover, tumor-associated microbes can influence local and systemic immunity by affecting antigen presentation, cytokine production, and immune cell infiltration, creating an immunosuppressive microenvironment that fosters tumor progression and therapeutic resistance (36).

The growing recognition that microbial ecosystems significantly influence cancer development, progression, and therapeutic response underscores the critical need for diagnostic tools that can capture this complex biological interplay alongside the tumor's own characteristics. Effectively monitoring cancer diseases, including potential microbial influences, requires moving beyond traditional methods. This clinical need has driven the rapid development and adoption of innovative approaches, prominently featuring LB.

## THE STRENGTH OF LB IN CANCER DIAGNOSTICS

LB represents a major advance, offering a minimally invasive window into tumor biology that overcomes many drawbacks of conventional techniques. Clinicians require a precise diagnosis that includes details about the tumor's biology and genetics to make confident cancer treatment decisions. Molecular fingerprints of the tumor are essential for monitoring how the cancer responds to therapy and tracking its progression over time. However, conventional tissue biopsies are invasive and provide only a limited, time-specific, and spatially restricted snapshot of the tumor. Moreover, certain tumor locations are particularly challenging to access. In addition, radiological imaging has its own drawbacks, including high costs, patient exposure to radiation, and limited molecular information.

In contrast, LB has emerged as a powerful, minimally invasive strategy for detecting, tracking, and treating cancer (37). A key focus—though not the only one—is the analysis of cell-free DNA (cfDNA), which mainly derives from the hematopoietic system, with smaller fractions from solid tissues (38). First reported by Mandel and Metais (39), cfDNA has been extensively studied across various medical fields, particularly in oncology. In cancer patients, tumor cells can release DNA fragments into the bloodstream, adding circulating tumor DNA (ctDNA) to the cfDNA pool (38). The ability to analyze these fragments via a simple blood draw has transformed ctDNA into a promising alternative to conventional biopsy and a widely used prognostic and predictive biomarker (see recent reviews [37, 40, 41]). LB unlocks a broad range of clinical possibilities for both advanced and early-stage cancer settings.

One of the central applications of LB for advanced-stage tumors, now implemented into clinical practice, is to guide treatment decisions by detecting actionable genetic alterations and resistance mutations (42). Since ctDNA is thought to correlate with tumor burden, quantitative changes in ctDNA levels over time have proven to be a valuable indicator of therapeutic efficacy (42–44). The non-invasive nature of ctDNA analysis enables more frequent assessments than tissue biopsies and imaging techniques, further underscoring its potential for tracking treatment response. More recently, the application of LB has expanded to earlier disease stages, where ctDNA-based approaches aim to detect minimal residual disease after curative treatments, enabling early relapse detection and informing adjuvant therapy decisions to adjust treatment intensity (45–48). Notably, intensive research is underway to apply ctDNA approaches for non-invasive cancer screening (49–52).

## CIRCULATING BIOMARKERS IN BODY FLUIDS

LB is an umbrella term describing the analysis of circulating biomarkers found in peripheral blood and other body fluids, such as cerebrospinal fluid, pleural fluid, peritoneal fluid, saliva, and urine (53). Traditionally, LB has focused on detecting

biomarkers of human self-origin—molecules released by the patient's own cells, whether healthy or diseased. These include cfDNA, ctDNA, circulating tumor cells, extracellular vesicles, proteins, and messenger and non-coding RNA (41) (Fig. 1A). Each biomarker, individually and in combination, provides valuable insights into physiological processes and disease states.

Beyond these self-derived biomarkers, LB is also applied to detect human biomarkers from different origins, such as fetal cfDNA in maternal circulation for prenatal screening (54) or donor-derived cfDNA in transplant patients for graft monitoring (55).

More recently, it has become evident that not only human cells release nucleic acids into the bloodstream (Fig. 1). A small and variable but biologically significant fraction of cfDNA—approximately <0.1% to 1%—originates from non-human sources (56–60), particularly from microbes colonizing or invading the host. Most hypotheses about the origin of this microbial DNA in the bloodstream are based on direct shedding from tumor sites (tumor necrosis), barrier translocation (leaky gut), microbial extracellular vesicles carrying DNA, transient bacteremia, host immune cell phagocytosis, and artifacts such as technical contamination (false positives) (56). Studies such as those by Tong et al. (61) showed that most of this mcfDNA comes from bacteria, while archaeal, fungal, and viral sequences are also detectable, with viral cfDNA offering particular diagnostic and prognostic utility in virally driven cancers (62). Although mcfDNA represents only a minor fraction of total cfDNA, its presence offers a unique opportunity to detect pathogenic microbes and explore the host-associated microbiome's biodiversity.

In essence, biomarkers derived from the host (e.g., ctDNA) and those from the associated microbiome (e.g., mcfDNA) represent distinct but complementary layers of biological information accessible through LB. Examining them in isolation, i.e., single-analyte approach, provides only a partial view. However, integrating these signals offers the potential for a far more comprehensive understanding of tumor biology, laying the foundation for enhanced diagnostics, treatment, and early detection strategies. This concept aligns with the growing shift toward a multi-view or multimodal LB, which seeks to combine multiple layers of information for a more complete picture of the disease (Fig. 1B). A thorough analysis of microbial cfDNA—capturing the diversity of bacteria, archaea, fungi, and viruses—and its integration with an expanded ctDNA profile that includes both genetic (single-nucleotide variants and copy number alterations) and nongenetic features (methylomics, fragmentomics, and nucleosomics), holds the potential to boost LB sensitivity significantly.

## EXPLORING ctDNA AND MICROBIAL FINGERPRINTS USING WGS

For the analysis of cfDNA, including both ctDNA and mcfDNA, two main methodological approaches are commonly used: PCR-based and NGS-based methods (63). PCR-based assays are highly sensitive but limited to detecting predefined targets. In contrast, NGS-based methods are more versatile and can be designed as targeted or untargeted approaches. Targeted NGS focuses on specific (pathogen) genes or regions. A notable example is the study by Chan et al. (64), where targeted sequencing of the entire Epstein-Barr virus (EBV) genome in plasma enabled more comprehensive and unbiased detection of EBV DNA, providing better risk stratification for nasopharyngeal carcinoma patients compared to conventional PCR approaches.

In contrast, untargeted methods such as WGS enable comprehensive profiling of nucleic acids within a sample, rather than focusing solely on specific regions. Although WGS shows promise for mcfDNA analysis, its implementation is challenged by the typically low abundance of microbial DNA in LB samples. This limitation reduces sensitivity and requires high sequencing depth, thereby increasing costs. Recent studies have explored enrichment strategies prior to sequencing since mcfDNA fragments tend to be shorter in size than host-derived cfDNA. One such approach achieved a remarkable 204-fold enrichment of the mcfDNA fraction by selectively removing longer host cfDNA fragments and utilizing single-stranded library methods to capture shorter fragments.

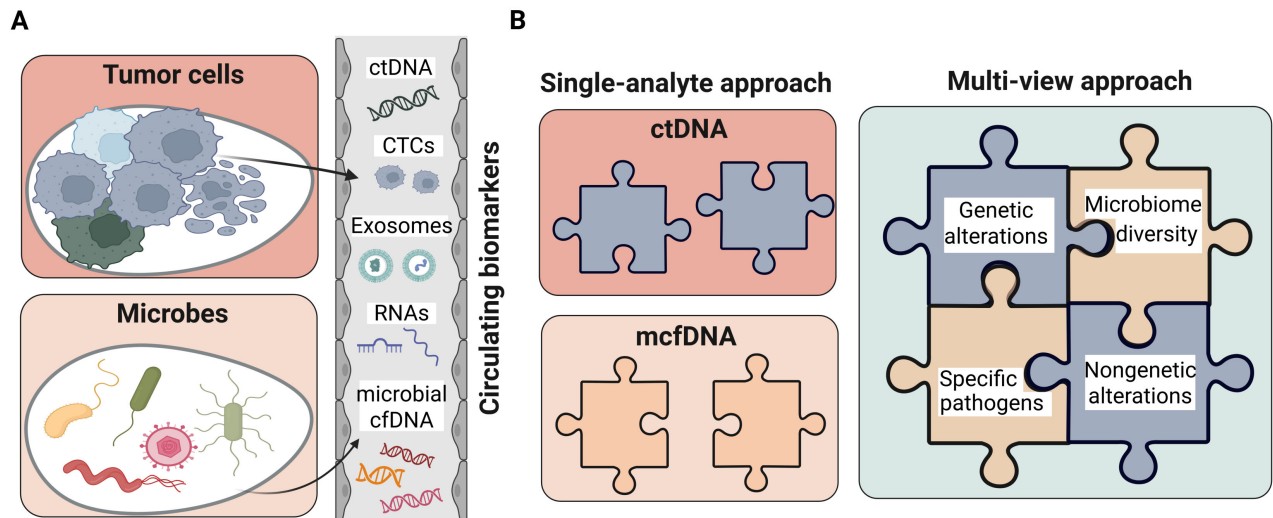

**FIG 1** (A) Schematic representation of potential circulating biomarkers from LB samples. (B) Single-analyte approaches examine either ctDNA or microbial cell-free DNA (mcfDNA) individually. The multi-view strategy integrates these distinct layers of information for a more comprehensive picture. For instance, a multi-view analysis might integrate genetic and nongenetic features of ctDNA alongside the total microbial diversity and specific pathogens from mcfDNA. Created in BioRender (E. Heitzer, 2025).

Nevertheless, this enrichment technique was reported to increase background noise at the genus level, potentially limiting sensitivity for pathogen detection (65).

However, the unique strength of untargeted WGS lies in its intrinsic capacity to simultaneously profile both host-derived cfDNA (including ctDNA) and non-human microbial DNA within a single LB sample. Most LB workflows are designed primarily for human nucleic acid analysis. Consequently, microbial sequences, although co-sequenced, are often removed during bioinformatic processing, wasting valuable microbial biomarker information (66). A study by Cheng and colleagues (67) highlighted the value of integrating both host and microbial information: by sequencing host urine cfDNA, they could detect tissue damage caused by microbial infections. Such a comprehensive approach expands the spectrum of potential biomarkers and provides deeper insights into tumor biology and its interplay with associated microbial fingerprints. Harnessing host and microbial signals from WGS could unlock new diagnostic and prognostic biomarkers, advancing LB toward a more integrated and insightful cancer assessment tool.

## UNIFICATION OF ctDNA AND MICROBIAL FINGERPRINT ANALYSES USING DATA FUSION APPROACHES

To realize the potential of combining cfDNA/ctDNA and microbial fingerprint analyses, robust data integration (data fusion) strategies are essential. The sheer volume and complexity of data generated, particularly from WGS, necessitate ML methods that can integrate multimodal inputs—i.e., different data types, such as host cfDNA/ctDNA and mcfDNA signals. Although dedicated host–microbe fusion is still an emerging area, established ML frameworks provide a strong starting point for investigation (68, 69).

Approaches to integrating these distinct biomarker types typically include early, late, and intermediate fusion strategies (Fig. 2).

## Early fusion

Early fusion is the simplest and most widely used data fusion strategy: it combines features from all modalities (see Box 1) into a single feature matrix. This enables the model to learn directly which combinations of features are most informative (69, 70). Early fusion performs best when the modalities contribute a comparable number of features, and these features are measured on similar numerical scales. To prevent one modality from dominating the model when feature numbers differ, dimensionality-reduction methods are often applied to reduce redundancy and balance the contribution of each modality (68). Early fusion is particularly appropriate when most features already carry high intrinsic predictive value, meaning they are informative on their own. For example, Medina et al. (71) developed an early detection model for ovarian cancer by linking cfDNA fragment length profiles with protein biomarkers using penalized logistic regression (see Box 1). Because the protein markers already have a high intrinsic predictive power, the cfDNA fragment length data (originally >400 bins) were first summarized as a short-to-long fragment ratio (<151 bp/>150 bp). Principal component analysis was then used to condense these ratios into two components further. These high-value cfDNA features were subsequently combined with the informative protein markers.

Additional feature reduction approaches, such as multidimensional scaling (65) or autoencoders (72), are also frequently employed to compress high-dimensional data into a smaller set of informative features (see Box 1).

The key principle of early fusion is that features from different modalities provide complementary information that the model can exploit directly. If the modalities require separate processing to become informative, then intermediate or late fusion strategies may be more suitable (69).

## Late fusion

Late fusion trains a separate model for each modality and then combines their predictions into a single output. Each modality thus has its own dedicated classifier or regressor (see Box 1), and the final prediction is obtained by combining these individual outputs—either by simple averaging or by using a meta-learner, i.e., a simple model that combines the predictions. In this setup, each model serves as an expert for its modality, contributing an independent assessment to the final decision. For example, Kim et al. (70) trained dedicated classifiers for cfDNA methylation, copy number, and fragmentation size to predict both cancer detection and tissue of origin (TOO). By averaging the support vector machine (SVM) classifier outputs across these three modalities (see Box 1), they improved sensitivity for cancer detection and TOO prediction. Similarly, pancreatic cancer detection was enhanced using a late fusion strategy in which fragment sizes were distilled into the DELFI score (52), and fragment end patterns served as surrogates for methylation signatures. A simple logistic regression meta-learner then combined both scores to generate the final prediction (73).

According to Ramachandram and Taylor (68), neither early nor late fusion is inherently superior; the optimal approach depends on the specific problem context. This observation was further supported by the SPOT-MAS model for multi-cancer detection and TOO prediction (74). In this study, both fusion techniques were evaluated using features such as 450 target methylation regions, genome-wide methylation, fragment size, copy number alterations, and fragment end motifs. The late fusion approach, which integrated modality-specific predictions via a meta-learner (see Box 1), notably improved discrimination between healthy and cancer samples. Conversely, the early fusion approach, implemented via a graph convolutional neural network (see Box 1), achieved superior accuracy for TOO prediction (74). These findings suggest that while late fusion excels at

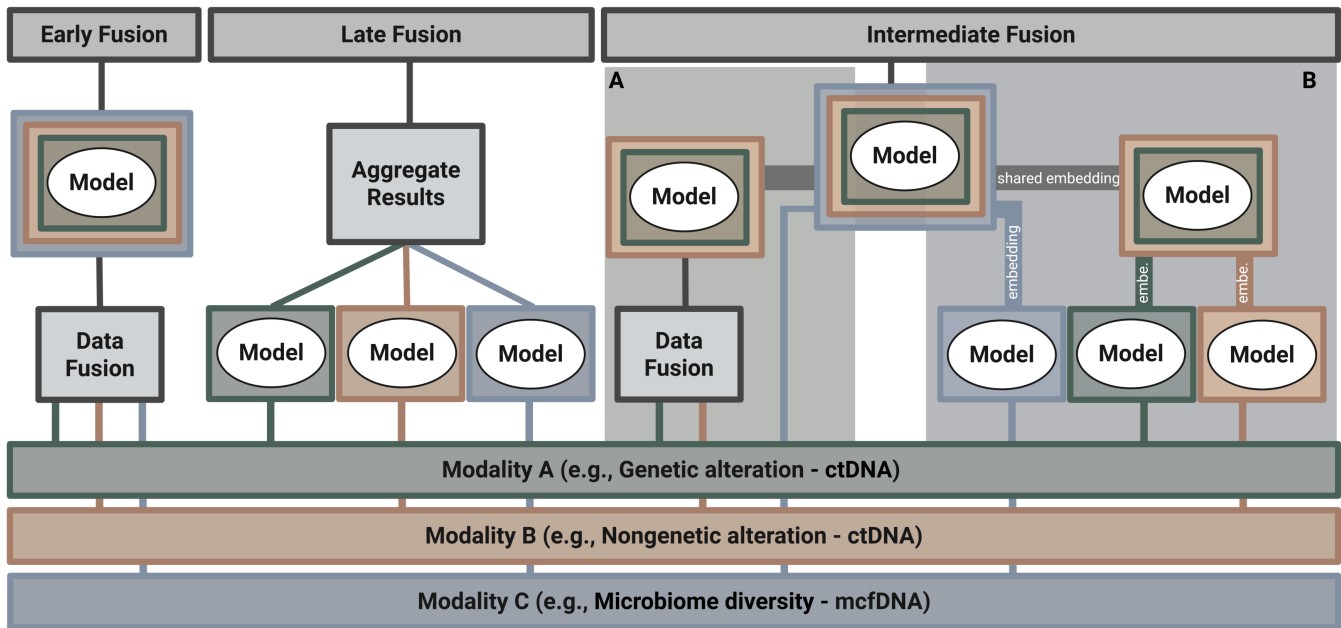

**FIG 2** Combining different modalities using early, late, and intermediate fusion. Each fusion approach has its advantages. While early fusion provides possible orthogonal information, late fusion techniques become experts in their own modality (e.g., genetic and nongenetic alterations of ctDNA and mcfDNA). Intermediate fusion serves as a flexible bridge between early and late fusion, combining features or representations in a stepwise manner. Variants include (A) fusion through a joint embedding integrated with an additional modality and (B) sequential fusion of modality-specific embeddings. Created in BioRender (E. Heitzer, 2025).

distinguishing healthy from cancer cases, it may fail to fully capture cross-modal relationships that are critical for more complex predictions, such as TOO classification.

## Intermediate fusion

Intermediate fusion aims to overcome the limitations of both early and late fusion. Early fusion may combine features too soon, before each modality's features are individually informative, while late fusion may overlook relationships between modalities by merging only final predictions. Intermediate fusion provides a stepwise strategy that allows gradual integration of information across modalities (68). It can be implemented as a slow fusion process, in which information is progressively integrated across multiple stages (75).

Intermediate fusion is a flexible framework that can be adapted to the degree of complementarity or overlap between modalities. When modalities contribute distinct yet complementary signals, they can be combined into a shared embedding—a compact representation that unifies the informative signals across modalities. This embedding can then be fused with additional data types or fed into a downstream classifier for prediction (Fig. 2A). For instance, genetic and nongenetic features of ctDNA capture similar underlying tumor-specific signals but reveal distinct patterns. Combining them into a shared embedding (e.g., via an autoencoder) helps the model learn connections between these related modalities before integrating more distinct but complementary information, such as mcfDNA, for prediction.

Alternatively, each modality can be distilled independently—using autoencoders, neural networks, or other feature-extraction models—to generate modality-specific embeddings (see Box 1). These embeddings are then combined stepwise into a joint representation that integrates cross-modal information for downstream prediction (Fig. 2B). While not yet applied in the LB field, this approach has been successfully used to diagnose Alzheimer's disease by integrating magnetic resonance imaging and positron emission tomography data using a stacked autoencoder. This joint representation was

then fed into an SVM for classification (76). Together, these strategies make intermediate fusion a flexible bridge between early and late fusion, preserving the level of complementarity or overlap across modalities.

Data integration strategies—early, intermediate, and late fusion—are key for combining ctDNA with microbial fingerprints. Because each method has distinct strengths, the choice of fusion strategy should be guided by the data characteristics and the underlying biological question.

## LIMITATIONS, DEVELOPMENTS, AND FUTURE OUTLOOK

While LB has the potential to extract microbial cfDNA, WGS analysis faces critical hurdles, including low sensitivity, specificity, and reproducibility. Foremost among these is the lack of standardized, validated methods for robustly characterizing the low-abundance mcfDNA component while accounting for the complex, dynamic, and personalized nature of the host-microbiome system. Specific technical challenges include mitigating pervasive contamination through rigorous controls and filtering and overcoming low microbial yields relative to host DNA using validated enrichment or depletion strategies. In addition, accurate bioinformatic classification must be ensured despite short reads and database limitations, requiring the use of stringent algorithms and curated reference resources. Furthermore, validating the significance of combined ctDNA-mcfDNA signatures and disentangling these signals from noise remains a significant research challenge.

Key developments addressing these issues include optimized laboratory protocols for co-analysis and potentially leveraging alternative sequencing technologies, such as real-time long-read sequencing, to improve microbial classification. Future priorities must therefore include (i) establishing standardized, validated end-to-end protocols and pipelines for the parallel analysis of ctDNA and mcfDNA; (ii) developing and rigorously evaluating novel, reliable, and clinically relevant biomarkers based on combined host and microbial signatures, demonstrating their reproducibility, predictive power, and utility compared to single-analyte approaches; (iii) conducting further studies—potentially including longitudinal, interventional, and relevant model system approaches—to elucidate causal links between specific microbial signatures and host cancer biology; and (iv) finally, unlocking the full potential of combined ctDNA–mcfDNA analysis will depend on advanced ML-based data fusion strategies and tailored bioinformatic approaches to capture all relevant molecular and microbial insights.

## ACKNOWLEDGMENTS

During the preparation of this work, the authors used ChatGPT for language editing. The authors carefully reviewed and revised the content and take full responsibility for the content of the publication.

## AUTHOR AFFILIATIONS

[1]Institute of Human Genetics, Diagnostic & Research Center for Molecular BioMedicine, Medical University of Graz, Graz, Austria
[2]Institute of Hygiene, Microbiology and Environmental Medicine, Diagnostic & Research Center for Molecular BioMedicine, Medical University of Graz, Graz, Austria

## AUTHOR ORCIDs

Alexander Mahnert  http://orcid.org/0000-0001-7083-8894

## FUNDING

| Funder | Grant(s) | Author(s) |
|---|---|---|
| Medizinische Universität Graz | ASO190003011 | Alexander Mahnert |

## AUTHOR CONTRIBUTIONS

Tina Moser, Conceptualization, Funding acquisition, Writing – original draft, Writing – review and editing | Matthias J. Moser, Writing – original draft, Writing – review and editing | Alexander Mahnert, Conceptualization, Funding acquisition, Project administration, Resources, Writing – original draft, Writing – review and editing

## ADDITIONAL FILES

The following material is available online.

### Open Peer Review

**PEER REVIEW HISTORY (review-history.pdf).** An accounting of the reviewer comments and feedback.

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
