## [Reviewer comments · mSystems]

The best from both disciplines - integrating human and microbial signatures from whole genome sequencing to advance cancer diagnostics

Tina Moser, Matthias Moser, and Alexander Mahnert

Corresponding Author(s): Alexander Mahnert, Medizinische Universitat Graz

Review Timeline:

Submission Date:	April 23, 2025
Editorial Decision:	September 11, 2025
Revision Received:	November 4, 2025
Accepted:	November 17, 2025

Editor: Pedro Oliveira

Reviewer(s): The reviewers have opted to remain anonymous.

Transaction Report:

DOI: <https://doi.org/10.1128/msystems.00039-24>

Re: mSystems00039-24 (**The best from both disciplines - integrating human and microbial signatures from whole genome sequencing to advance cancer diagnostics**)

Dear Dr. Mahnert:

My apologies for the delay, as it was extremely difficult to find suitable reviewers. Below you will find instructions from the mSystems editorial office, and the reviewer comments.

Revision Guidelines

- Upload point-by-point responses to the issues raised by the reviewers in a file named "Response to Reviewers," NOT in your cover letter.
- Upload a compare copy of the manuscript (without figures) as a "Marked-Up Manuscript" file.
- Upload a clean .DOC/.DOCX version of the revised manuscript and remove the previous version.
- Each figure must be uploaded as a separate, editable, high-resolution file (TIFF or EPS preferred), and any multipanel figures must be assembled into one file.

Minireviews are not subject to publication charges.

Author Bios: We encourage you to submit a biographical sketch of each author (limit of 150 words) along with a photo to be published at the end of your article. You can submit these with your modified manuscript.

Figures Enhancement: ASM has engaged a professional science illustrator, Patrick Lane of ScEYence Studios, to work with minireview authors at the modification stage to generate improved figures that are uniform throughout the journal. This art enhancement service is free of charge to authors of minireviews and full-length reviews, and turnaround time is fast. I think you will be pleased with the results. Please contact Patrick on receiving this letter. Complete contact information for Patrick and further instructions are posted at <https://journals.asm.org/pb-assets/pdf-text-excel-files/graphical-enhancement-support.pdf>.

Sincerely,
Pedro Oliveira
Editor
mSystems

Reviewer #1 (Comments for the Author):

Dear Editor,

I have reviewed the manuscript submitted to mSystems by Moser et al. and entitled "The best from both disciplines - integrating human and microbial signatures from whole genome sequencing to advance cancer diagnostics".

This mini-review addresses a very interesting point, that is the benefit of considering fractions of liquid biopsies during oncology diagnostics that are typically filtered out, namely the DNA included in microbial cells that is typically discarded when focusing on cell-free DNA. Authors present convincing arguments for inclusion of this and some guidance on the various ways to implement ML approaches to do it.

General comment:

I don't have many comments and I am generally happy about the manuscript. My main criticism would be that the various fusions strategies are explained in a very technical terms or ML jargon, perhaps consider simplifying it for the non-ML using

more general or layman's terms. For instance, it would be good to explain what exactly are "modalities", "multimodalities". For instance, the sentences below are all very unclear to non-ML scientists. It's all understandable if you are used to ML jargon, but slightly more examples and explanations on what it exactly meant would be very beneficial.

- "reduce unintended redundancies in the input vector" (lines 220-221)
- "However, the main idea of early fusion is that we expect features in different modalities to have hidden orthogonal information. Suppose this is not the case, and features of different modalities can become informative only in a post processed step, other fusion strategies are necessary" (lines 231-234)
- "distilling modules" (line 237)
- "additional meta-learner" (line 239)
- Etc.
-

Minor comments:

- General comment, prompted by lines 16-20 and lines 154-156: Would microbes (and DNA) found in blood be particularly relevant to tumours or can they be there for other reasons? Are there hypotheses as to why/how this DNA of microbial origin is found in the circulatory system?
- Line 12: a brief explainer to why?
- Line 29: till until
- Line 72: "Most of the human microbiome is located in the GI tract"; in terms of biomass? Diversity? Specify.
- Line 157: about pathogenic microbes. Is this realistic or only theoretical? This must be an even lower fraction of the low detectable fraction...
- Line 230: Autoencoders is wrongly capitalized

Reviewer #2 (Comments for the Author):

The mini-review provides an overview of integrating ctDNA and mcfDNA signals via whole-genome sequencing and machine learning data fusion for enhanced cancer diagnostics.

I have just a few suggestions, that I hope will help improving the manuscript.

- 1) The manuscript states that mcfDNA represents "approximately 1%" of total cfDNA, citing Huang et al 2018 and Kowarsky et al. 2017. While these early studies reported ~1% nonhuman sequences in cfDNA (with a subset being microbial), more recent analyses indicate that true microbial cfDNA is often much lower, typically <0.1-1% and highly variable. My suggestion would be to revise this value, noting that it can vary by sample and method, adding appropriate citations.
- 2) The manuscript discusses the controversy between Poore et al 2020 on cancer-specific microbial signatures, Gihawi et al (2023) critiquing methodological flaws, and Sepich-Poore et al 2024 defending robustness with revised workflows. However, the original Poore et al. Nature paper was retracted in 2024 due to concerns over the reliability of microbial-cancer associations (Nature retraction note here <https://www.nature.com/articles/s41586-024-07656-x>). I think the authors should mention this retraction or remove this citation.
- 3) The authors cite Battaglia et al 2024 for Fusobacterium in lung cancer resistance to immune checkpoint blockade. However, recent papers highlight broader mechanisms, such as intratumoral bacteria modulating DNA damage, epigenetic changes, and immune evasion across cancers. I would add 1-2 sentences with examples on this.
- 4) I would slightly revise the sentence "most of this microbial cell-free DNA (mcfDNA) comes from bacteria, while archaeal, fungal and viral sequences are also detectable" to include, for example: "...with viral mcfDNA offering specific diagnostic utility in virally driven cancers like nasopharyngeal carcinoma" (or similar). And I would add references such as Chan et al., 2022, Ann Oncol.
- 5) The data fusion section provides good general examples, but seems to lack direct applications to mcfDNA-ctDNA integration. Recent work used ML for circulating microbiome DNA in early detection, achieving high accuracy by fusing methylation and microbial profiles (for example Li et al., 2024, <https://www.sciencedirect.com/science/article/pii/S266637912400380X>). I would add a paragraph or example stating that intermediate fusion via autoencoders could distill microbial diversity features before integrating with ctDNA fragmentation.

Response to Reviewers of

“The best from both disciplines - integrating human and microbial signatures from whole genome sequencing to advance cancer diagnostics” (mSystems00039-24)

REVIEWER’S COMMENTS

Reviewer #1 (Comments for the Author):

Dear Editor,

I have reviewed the manuscript submitted to mSystems by Moser et al. and entitled "The best from both disciplines - integrating human and microbial signatures from whole genome sequencing to advance cancer diagnostics".

This mini-review addresses a very interesting point, that is the benefit of considering fractions of liquid biopsies during oncology diagnostics that are typically filtered out, namely the DNA included in microbial cells that is typically discarded when focusing on cell-free DNA. Authors present convincing arguments for inclusion of this and some guidance on the various ways to implement ML approaches to do it.

Response: We thank the reviewer for the thoughtful evaluation of our manuscript and for highlighting the conceptual motivation of the review. We appreciate the recognition of our central aim: to argue that microbial DNA signals, which are often discarded in conventional liquid biopsy workflows, can provide complementary diagnostic value alongside host-derived cfDNA. We are also grateful for the positive assessment of how machine learning strategies can be used to integrate these signals in practice.

General comment:

Comment 1: I don't have many comments and I am generally happy about the manuscript. My main criticism would be that the various fusions strategies are explained in a very technical terms or ML jargon, perhaps consider simplifying it for the non-ML using more general or layman's terms. For instance, it would be good to explain what exactly are "modalities", "multimodalities".

For instance, the sentences below are all very unclear to non-ML scientists. It's all understandable if you are used to ML jargon, but slightly more examples and explanations on what it exactly meant would be very beneficial.

Response: We thank the reviewer for this constructive comment and fully agree that several sections, in particular the descriptions of the fusion strategies, were initially written in language that presumes familiarity with machine learning terminology. Our intention is for this review to be accessible not only to computational researchers but also to clinicians, molecular biologists, and translational researchers. We therefore revised the text throughout the “Unification of ctDNA and microbial fingerprint analyses using data fusion approaches” sections to reduce jargon and explain key terms in plain language.

In addition, we now provide brief explanatory definitions at first use and include a short Glossary. We explicitly define the terms *feature*, *modality*, *multimodal*, *autoencoder*, *penalized logistic regression*, *multidimensional scaling (MDS)*, *support vector machine (SVM)*, *meta-learner*, *classifier*, *regressor*, *neural networks*, and *graph convolutional neural network (GCN)* (Glossary, lines 364–412), so that non-ML readers can follow the remainder of the section.

We carefully revised the specific passages the reviewer highlighted.

- "reduce unintended redundancies in the input vector" (lines 220-221)

Revised text: *"To prevent one modality from dominating the model when feature numbers differ, dimensionality-reduction methods are often applied to reduce redundancy and balance the contribution of each modality (68)."* (lines 238-240)

- "However, the main idea of early fusion is that we expect features in different modalities to have hidden orthogonal information. Suppose this is not the case, and features of different modalities can become informative only in a post processed step, other fusion strategies are necessary" (lines 231-234)

Revised text: *"The key principle of early fusion is that features from different modalities provide complementary information that the model can exploit/combine directly. If the modalities require separate processing to become informative, then intermediate or late fusion strategies may be more suitable (69)."* (lines 253-256)

- "distilling modules" (line 237)

We have rewritten the whole paragraph.

- "additional meta-learner" (line 239)

We included the term "meta-learner" in the Glossary for completeness.

- Etc.
-

Overall, we have substantially revised the fusion strategy section (early, intermediate, and late fusion) to clarify when each approach is appropriate and minimize machine learning jargon. We believe the text is now more accessible to a broad range of readers.

Minor comments:

Comment 2: - General comment, prompted by lines 16-20 and lines 154-156: Would microbes (and DNA) found in blood be particularly relevant to tumours or can they be there

for other reasons? Are there hypotheses as to why/how this DNA of microbial origin is found in the circulatory system?

Response: We thank the reviewer for raising this important point. We agree that microbial DNA detected in blood is not exclusively tumor-derived, and that possible sources and mechanisms should be made explicit in the manuscript. We have now added a short explanatory paragraph to the text (lines 166-170) that summarizes current hypotheses regarding the origin of circulating microbial DNA.

Of course, there are numerous other sources of microbial DNA signatures in the circulatory system. We have added some common hypotheses and a recent reference.

“Most hypotheses about the origin of this microbial DNA in the bloodstream are based on direct shedding from tumor sites (tumor necrosis), barrier translocation (leaky gut), microbial extracellular vesicles carrying DNA, transient bacteremia, host immune cell phagocytosis, but also artifacts like technical contaminations (false positives) (Zhang et. al, 2025).” (lines 166-170)

Zhang, Z., Ren, W., Wang, Y., Zhai, T., & Huang, J. (2025). The origin of circulating microbial DNA in the blood: where does it come from? Annals of Medicine, 57(1). <https://doi.org/10.1080/07853890.2025.2560605>

Comment 3: - Line 12: a brief explainer to why?

Response: *Thanks for the hint. We have revised the opening sentence of the manuscript not only to state that liquid biopsies are transforming oncology, but also to briefly explain why they are impactful in clinical practice.*

We revised the sentence to include some information about why liquid biopsies transform oncology.

“Liquid biopsies are transforming oncology enabling earlier diagnosis, dynamic treatment guidance, and personalized precision medicine, yet current approaches focusing mainly on circulating host cell-free DNA (cfDNA) neglect crucial information within co-existing microbial cell-free DNA (mcfDNA).” (lines 12-15)

Comment 4: - Line 29: till ◊ until

Response: *Corrected.*

Comment 5: - Line 72: "Most of the human microbiome is located in the GI tract"; in terms of biomass? Diversity? Specify.

Response: We appreciate this clarification request. We specified this phrase in terms of biomass (as diversity could be highly variable).

“Most of the human microbiome’s biomass is located in the gastrointestinal tract, especially in the colon, i.e., the gut microbiome”. (lines 74-75)

Comment 6: - Line 157: about pathogenic microbes. Is this realistic or only theoretical? This must be an even lower fraction of the low detectable fraction...

Response: We consider this theoretical, but expect rather high proportions e.g., during bacteremia.

Comment 7: - Line 230: Autoencoders is wrongly capitalized

Response: We thank the reviewer for pointing this out. We have corrected it.

Reviewer #2 (Comments for the Author):

The mini-review provides an overview of integrating ctDNA and mcfDNA signals via whole-genome sequencing and machine learning data fusion for enhanced cancer diagnostics.

I have just a few suggestions, that I hope will help improving the manuscript.

Response: We appreciate the reviewer's constructive suggestions for further improving the clarity of our mini-review, and we have addressed each point in detail below.

Comment 1: The manuscript states that mcfDNA represents "approximately 1%" of total cfDNA, citing Huang et al 2018 and Kowarsky et al. 2017. While these early studies reported ~1% nonhuman sequences in cfDNA (with a subset being microbial), more recent analyses indicate that true microbial cfDNA is often much lower, typically <0.1-1% and highly variable. My suggestion would be to revise this value, noting that it can vary by sample and method, adding appropriate citations.

Response: We thank the reviewer for this valuable comment. We agree that more recent datasets suggest that the fraction of microbial cfDNA in plasma is often well below 1% of total cfDNA, and that this fraction is highly variable across individuals, clinical states, and analytical pipelines. We re-examined the current literature and identified recent studies reporting that microbial cfDNA typically constitutes a very small but detectable fraction (often <0.1–1% of total cfDNA), and that technical factors (e.g., background contamination, library preparation, bioinformatic filtering) strongly influence these estimates. We have updated the text accordingly and added citations.

"A small and variable but biologically significant fraction of cfDNA — approximately <0.1 - 1%— originates from non-human sources". (lines 164-165)

We have added the following supporting references in this context:

doi: 10.1080/07853890.2025.2560605

doi: 10.1128/msystems.00008-24

doi: 10.3389/fcimb.2024.1458316

Comment 2: The manuscript discusses the controversy between Poore et al 2020 on cancer-specific microbial signatures, Gihawi et al (2023) critiquing methodological flaws, and Sepich-Poore et al 2024 defending robustness with revised workflows. However, the original Poore et al. Nature paper was retracted in 2024 due to concerns over the reliability of microbial-cancer associations (Nature retraction note here <https://www.nature.com/articles/s41586-024-07656-x>). I think the authors should

mention this retraction or remove this citation.

Response: *We agree, but we do not want to take either side and wish to remain as objective as possible. We have therefore amended the sentence accordingly.*

“Meanwhile, Sepich-Poore et al. retracted their original manuscript from 2020, but reaffirmed the robustness of cancer-associated microbiome signals using updated bioinformatic workflows that included batch correction, improved host read depletion with a newer human genome reference, cross-method consistency analyses, and additional validation steps”. (lines 50-54)

Comment 3: The authors cite Battaglia et al 2024 for Fusobacterium in lung cancer resistance to immune checkpoint blockade. However, recent papers highlight broader mechanisms, such as intratumoral bacteria modulating DNA damage, epigenetic changes, and immune evasion across cancers. I would add 1-2 sentences with examples on this.

Response: *Actually, this would be a good fit here. We added the following information and references:*

“Furthermore, intra-tumoral bacteria can influence tumor biology through multiple, conserved mechanisms across cancer types. For instance, certain bacterial genotoxins, such as colibactin produced by Escherichia coli and cytolethal distending toxin from Campylobacter spp., can directly induce DNA double-strand breaks and genomic instability, thereby promoting mutagenesis and tumor evolution (Tekle and Garrett, 2023). Bacterial metabolites and cell wall components can also alter chromatin accessibility, modulate histone modifications, and reshape DNA methylation patterns, ultimately reprogramming oncogenic signaling and stress-response pathways (Tekle and Garrett, 2023; Yang et al., 2023). Moreover, tumor-associated microbes can influence local and systemic immunity by affecting antigen presentation, cytokine production, and immune cell infiltration, creating an immunosuppressive microenvironment that fosters tumor progression and therapeutic resistance (Yang et al., 2023)” (lines 102-112)

Doi: 10.1038/s41568-023-00594-2

Doi: 10.1038/s41392-022-01304-4

Comment 4: I would slightly revise the sentence "most of this microbial cell-free DNA (mcfDNA) comes from bacteria, while archaeal, fungal and viral sequences are also detectable" to include, for example: "...with viral mcfDNA offering specific diagnostic utility in virally driven cancers like nasopharyngeal carcinoma" (or similar). And I would add references such as Chan et al., 2022, Ann Oncol.

Response: *We slightly revised the sentence as suggested.*

“...most of this microbial cell-free DNA (mcfDNA) comes from bacteria, while archaeal, fungal, and viral sequences are also detectable, with viral cfDNA offering particular diagnostic and prognostic utility in virally driven cancers.” (lines 170-173)

Comment 5: The data fusion section provides good general examples, but seems to lack direct applications to mcfDNA-ctDNA integration. Recent work used ML for circulating microbiome DNA in early detection, achieving high accuracy by fusing methylation and microbial profiles (for example Li et al., 2024, <https://www.sciencedirect.com/science/article/pii/S266637912400380X>). I would add a paragraph or example stating that intermediate fusion via autoencoders could distill microbial diversity features before integrating with ctDNA fragmentation.

Response: We agree that an explicit discussion of how to integrate microbial cfDNA and host-derived cfDNA/ctDNA features is essential. The field of multimodal fusion between ctDNA and mcfDNA is still in an early phase. Only a few studies have attempted to combine microbial signals with host cfDNA-derived features. For example, Zhou et al. (PMID: 40298367) presented a proof-of-concept study in colorectal cancer that combined blood-derived microbial features with cfDNA fragmentation features (i.e., fragment length) to build diagnostic classifiers. In that study, microbial signatures were first distilled, and these selected microbial features were then combined with cfDNA fragment length features using an early fusion strategy to train a final random forest classifier. The authors reported improved performance when using both data types together compared with using microbiome features alone.

However, we did not include this study in our review as a best-practice example due to concerns about the rigor of the machine learning technique. In particular, microbial feature selection (performed with AU-CRF) and model training were applied to the entire dataset before splitting into training and validation sets. Performing feature selection across all samples before creating a held-out test set can lead to information leakage from the validation set into the training process, artificially inflating performance estimates and limiting the generalizability of the reported accuracy. For this reason, in the manuscript, we were careful not to present this approach as an established or validated fusion pipeline, even though it is an important early demonstration that microbial and fragmentomic signals can, in principle, be combined.

Instead, we chose to frame fusion strategies as methodologically robust approaches that can be directly adopted to integrate microbial signatures with host cfDNA/ctDNA in prospective studies.

In line with the reviewer's suggestion, we have now added a concrete example of how an intermediate fusion strategy could be applied to ctDNA + mcfDNA in a rigorous way (lines 297-300).

“For instance, genetic and nongenetic features of ctDNA capture similar underlying tumor-specific signals but reveal distinct patterns. Combining them into a shared embedding (e.g., via an autoencoder) helps the model learn connections between these related modalities before integrating more distinct but complementary information, such as mcfDNA, for prediction”

Finally, regarding the Li et al. (2024) study cited in the comment: this work uses a transformer-based model to detect early ovarian cancer from cfDNA methylation patterns. It focuses on host-derived cfDNA methylation and does not integrate microbial cfDNA features. It represents a powerful single-modality methylation model rather than a ctDNA–mcfDNA fusion approach.

Re: mSystems00039-24R1 (**The best from both disciplines - integrating human and microbial signatures from whole genome sequencing to advance cancer diagnostics**)

Dear Dr. Alexander Mahnert:

Your manuscript has been accepted, and I am forwarding it to the ASM production staff for publication. Your paper will first be checked to make sure all elements meet the technical requirements. ASM staff will contact you if anything needs to be revised before copyediting and production can begin. Otherwise, you will be notified when your proofs are ready to be viewed.

Sincerely,
Pedro Oliveira
Editor
mSystems

Reviewer #2 (Comments for the Author):

The Authors have addressed all the points raised.